# Bidirectional Model Reconciliation: Explanations in Human-Robot Teams

## Abstract

As the use of AI technology becomes ubiquitous in our day-to-day lives, the need for AI agents to be either explicable or provide explanations (when they are not explicable) becomes important. An array of research work done in the automated planning community explores a human-robot interaction setting where the robot makes a plan and the human observes it and is provided with a set of explanations when the robot's plan (or behavior) does not align with the human's expectation. In this paper, we model a co-habitation scenario and provide a general overview where the human may either be a supervisor or a teammate. First, we highlight the various models that come into play in such settings. Second, we pin-point explanation scenarios that arise due to the disparity between the human and the robot about the understanding of team models. In settings where the robot is assumed to know more about the model, explanations are just a one-way communication. On the contrary, when the human's model is more accurate than that of the robot, we show that a two-way interaction becomes necessary for explanations. Lastly, we discuss how some of the existing works for the case where the human is a supervisor can be adapted in some of the settings when the human is a teammate and talk about a few high-level ideas that might help in scenarios for which no solutions exist.

There has been a renewed interest in the automated planning community for generating explanations in human-in-the-loop planning scenarios when a robot comes up with a plan that is incomprehensible to the human observer [1]. To this extent, researchers have shown that explanations can be viewed as a model reconciliation process that is highly effective in real-world interactions with humans [2]. In order to identify key differences between the human's understanding of a model and the actual model used by a robot, an array of work exists that can find a set of explanations in the context of a given plan [3, 4, 5]. These works primarily differ in the assumptions they make about the robot's idea about the human's understanding of its model (denoted as $((\mathcal{M}^R)_h)_r$).

We argue that human-robot teaming scenarios represent a more general setting and existing works on generating explanations (that do model reconciliation) is merely the tip of the ice-berg. Based on the role of the human in the team

setting, we can categorize the interaction between a human and a robot into two major categories– (1) the human is a supervisor and (2) the human is a fellow teammate capable of performing actions in the environment. In each setting, we then talk about scenarios where (1) the robot has a correct model of the world and good estimates of the human's models that is can use to come up with explanations, and (2) the robot's model (of the world and/or the human's models) are incorrect or imprecise in some way and thus, explanations generated might be incorrect and thereby, result in the start of an interaction that helps both the human and robot can update their models. This categorization is captured in Figure 1. We discuss how existing works have tried to address some of the challenges that arise when the human is a supervisor and if they can be adapted to settings where the human is a fellow teammate. Lastly, we discuss, at a relatively high-level, how we can use different forms of interaction to deal with cases when assumptions about human's model are incorrect or imprecise.

## Background

In this section, we formalize the notion of a model that we will use throughout the paper to highlight the various use cases for explanations. A model can be represented by the tuple $\mathcal{M}^\phi = \langle \mathcal{F}^\phi, \mathcal{A}^\phi, s_0, \mathcal{G} \rangle$, where $\mathcal{F}$ are the fluents that are used to define the state of the environment, $\mathcal{A}$ are the actions that an agent can perform in the environment, $s_0 \subseteq F$ is the initial state, which are the initial values of the fluents, $\mathcal{G}$ is the goal and $\phi$ denotes the agent whose model is being represented. Note that $\mathcal{F}^\phi$ are the fluents known to the agent $\phi$ and similarly, $\mathcal{A}^\phi$ is the set of actions $\phi$ can perform.

Every action $a \in \mathcal{A}$ consists of $\langle c(a), pre(a), eff^\pm(a) \rangle$ where, $c(a)$ is the cost associated with executing the action $a$ and $pre(a), eff^\pm(a) \subseteq \mathcal{F}$ are preconditions and add/delete effects of $a$. The transition function $\delta(s, a)$, represents the act of taking an action $a$ in state $s$, where, $s \subseteq \mathcal{F}$, and the state satisfies the preconditions for the actions, i.e. $\delta(s, a) \models \perp$ if $s \not\models pre(a)$. $\delta(s, a) = s'$, where $s' \subseteq \mathcal{F}$ and $s' = s \cup eff^+(a)/eff^-(a)$. A plan $\pi$ is a sequence of actions $\langle a_0, a_1, a_2, ..., a_n \rangle$. A plan is called valid if $\delta(s_0, \langle a_0, a_1, a_2, ..., a_n \rangle) \models \mathcal{G}$. Total cost of a plan $C(\pi) = \sum_{i=0..n} c(a_i)$. An optimal plan is $\pi^* = \arg\min_i C(\pi_i)$.

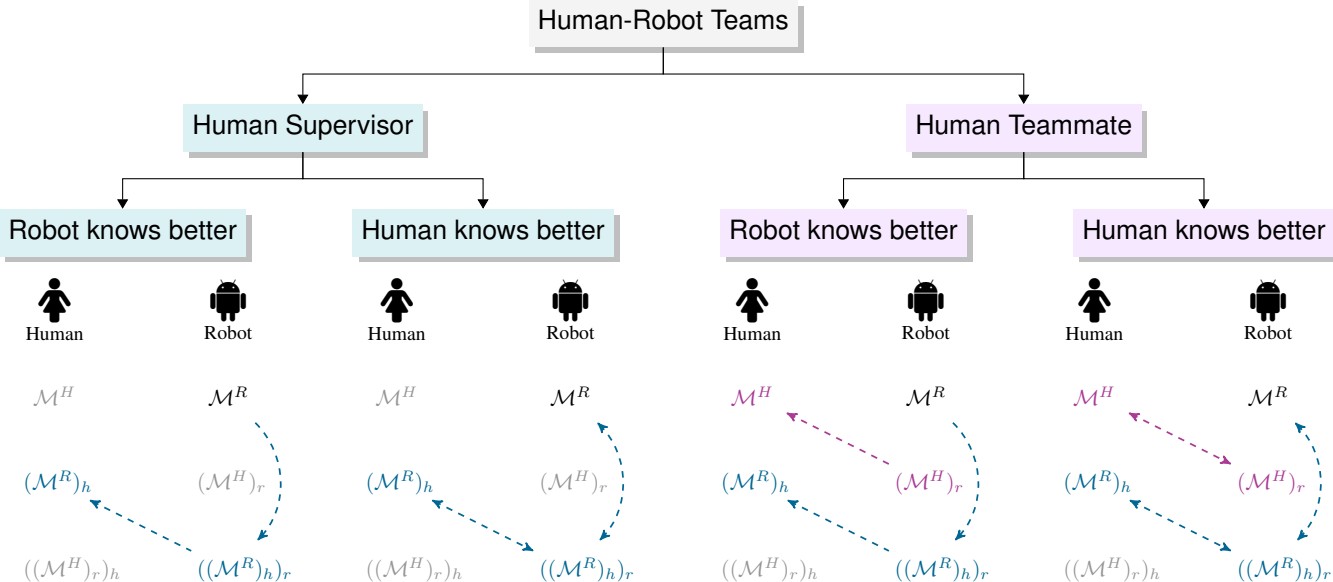

Figure 1: The various scenarios in human-robot teams. The phrase *robot knows better* indicates that its model about the world is accurate and its assumptions about the human's model is close the human's actual model. In these settings, one way communications suffice. In the setting where we say *human knows better*, we assume that the robot has an incorrect or incomplete model about the human's model. In this case, two-way communication is essential to explain a plan and re-plan if needed.

Additionally, we define $(\mathcal{M})_\phi$ as an approximation operator that represents the understanding of a model $\mathcal{M}$ by an agent $\phi$. For e.g., $(\mathcal{M}^H)_r$ is the robot's approximation of the human's model. Note that $(\mathcal{M}^\phi)_\phi = \mathcal{M}^\phi$, i.e. approximation of an agent's own model by themselves is their model.

**Human-In-the-Loop Planning (HILP)** In [6], the authors highlight the challenges that arise in automated planning when the human is no longer just a part of the environment but is interacting (in various ways) with the automated agent. Generating explicable or transparent behavior becomes a key goal for effective collaboration. When generating such behavior becomes expensive, the robot may choose to generate explanations that can help the human understand the plan at hand and/or correct it if the human realizes it is not. Particularly, the human can either be (1) a supervisor that observes and comments on the robot's plan or (2) a fellow teammate that also executes actions in the world alongside the robot. To understand the technicalities that arise in these scenarios, we use the approximation operator to highlight the six models that come into play in human robot settings. First, we highlight the three models that are sufficient to talk about the setting in which the human is a supervisor and heavily used in previous works [3, 5].

- $\mathcal{M}^R$ – mental model of the robot.
- $(\mathcal{M}^H)_r$ – robot's approximation of the human's model.
- $((\mathcal{M}^R)_h)_r$ – robot's approximation of the human's approximation or the robot's model.

In this setting, a plan $\pi = \langle a_0, a_1, a_2, \dots a_n \rangle$ is made by the robot and observed by the human supervisor and an explanation $\mathcal{E} = \langle \Delta R, \pi \rangle$ that is a set of model differences

$\Delta R$ given by $R$ to $H$ which ensures that the plan $\pi$ is valid and/or optimal in $((\mathcal{M}^R)_h)_r$. Second, we highlight the three other models that come into play when the human is a fellow teammate–

- $\mathcal{M}^H$ – mental model of the human.
- $(\mathcal{M}^R)_h$ – human's approximation of the robot's model.
- $((\mathcal{M}^H)_r)_h$ – human's approximation of the robot's approximation of the human's model.

In this case, the plan $\pi$ for the joint team is given by $\pi = \langle a_0^R, a_1^H, a_2^H, \dots a_n^R \rangle$, where the superscript denotes the agent who needs to perform the action. In this example, $a_0$ and $a_n$ are to be performed by the robot and $a_1$ and $a_2$ are to be performed by the human. Explanations here are represented by $\mathcal{E} = \langle \Delta R, \Delta H \rangle$ with the difference that $\Delta R$ and $\Delta H$ may comprise of model updates that relate to the understanding about the human's task model and the interactions between the human's and the robot's task model (maybe at some level of abstraction such as human will provide some resource that the robot needs or vice-versa) as opposed to just the robot's task model. We now look at the various use cases that arise in human-robot team settings.

## Use Cases

We present four use-cases based on the categorization shown in Fig. 1. We use an Urban Search and Rescue (USAR) scenario [7] to showcase specific examples for each use-case. In this human-robot team, we have a single human and a single robot agent tasked with the duty of finding and reporting locations of injured humans in a building that is on fire (refer figure 2). We will now categorize based on whether the

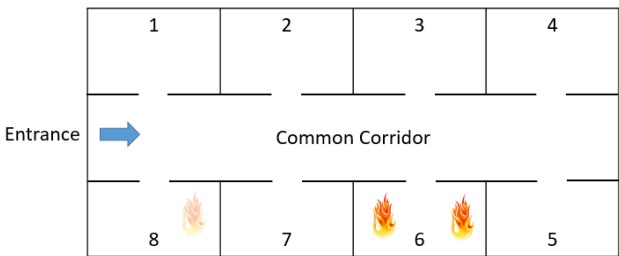

Figure 2: Urban Search and Rescue scenario with 8 rooms. Room 6 and Room 8 are on fire.

human is a supervisor or a fellow teammate that executes search and rescue operations alongside the robot.

## Human Supervisor

In this setting, the human acts as the supervisor to whom the robot is accountable and the robot's goal is to check all the rooms and report the exact locations of injured humans in the building. We further divide this case into two sub-cases based on who knows more about the domain (Fig. 1).

**Robot knows better**    In this case, the robot has (1) an accurate model of the domain, i.e. $\mathcal{M}^R$ is correct and the human's model (in the robot's mind), i.e. $((\mathcal{M}^R)_h)_r$ might be updated to explain a plan that was made using $\mathcal{M}^R$.
◇ **Use case** – Human has a map of the building and was expecting a plan in which the robot visits room 8 after visiting room 7. Unknown to them, due to fire in the building, a wall has fallen leaving room 7 inaccessible and the robot is aware of this. Thus, the robot comes up with a plan to search room 8 first and then use a door between room 8 and room 7 to check room 7. This plan is confusing for the supervisor and thus, there is a need to provide an explanation describing this change, which updates $((\mathcal{M}^R)_h)_r$ saying that entry to room 7 via the corridor is blocked. Note that, in this case, an explicable plan [8] might not be executable in the real world and thus, one has to resort to explanations.
◇ **Challenges** – There are two important challenges in this scenario (1) obtaining the model $((\mathcal{M}^R)_h)_r$ that is a good approximation of $(\mathcal{M}^R)_h$ and (2) finding explanations that reconcile $\mathcal{M}^R$ and $((\mathcal{M}^R)_h)_r$ in an efficient manner (in terms of reducing communication cost, while ensuring that the given plan is valid and/or optimal to the human).

There has been some work in this scenario where the authors in [3], performed model space search starting from $((\mathcal{M}^R)_h)_r$, assuming it to be available and a good approximation of $(\mathcal{M}^R)_h$), and then finding the model where the robot's plan is optimal by doing single predicate changes to the model. In [9] authors assumed that $((\mathcal{M}^R)_h)_r$ belongs to a set of models and use an annotated PDDL [10] to represent it efficiently. They assume that $(\mathcal{M}^R)_h$ is a part of this set. In [4] authors assumed that humans model is at a different level of abstraction and can be efficiently represented by dropping certain precondition and effects of actions.

Most of these works make the assumption that the robot will convince the human that this is the best plan to do by giving a set of model changes (which is the explanation).

This assumption falls flat when the human actual model of the robot $(\mathcal{M}^R)_h$ is different from $((\mathcal{M}^R)_h)_r$, the model the robot used to come up with explanations. In such cases, the human may reject some of the explanations and ask the robot to update its model. This brings us to the next use case.

**Human knows better**    When the human has a more accurate model of the the environment and the robot is unaware of it, the reconciliation process needs to happen in the other direction, i.e. the human with $(\mathcal{M}^R)_h$ can provide respnses to the robot that update both $((\mathcal{M}^R)_h)_r$ and $\mathcal{M}^R$.
◇ **Use case** – Continuing with the USAR example, let's assume human has the information about the fallen wall blocking room 7 whereas, the robot is not aware of it. In this case, the robot will come up with a plan to visit room 7 after room 6 and the human, via some form of interaction, will have to ensure that the robot's task model $\mathcal{M}^R$ is updated. This might need the human to provide model changes that makes the robot's plan ($\pi_{\mathcal{M}^R}$) infeasible in the current model $((\mathcal{M}^R)_h)_r$, which in turn updates $\mathcal{M}^R$. Finally, the robot might need to make another plan, leading to a classic case of replanning even before the execution phase starts.
◇ **Challenges** – The primary challenge here is to perform model reconciliation without the source model, i.e. to reconcile $((\mathcal{M}^R)_h)_r$ and $(\mathcal{M}^R)_h$. Since making the assumption $(\mathcal{M}^R)_h = ((\mathcal{M}^R)_h)_r$ is often a strong one, the robot can interact with the human to improve its estimate of the human's model, i.e make $((\mathcal{M}^R)_h)_r$ closer to $(\mathcal{M}^R)_h$.

Note that the robot cannot ask a human to simply enumerate $(\mathcal{M}^R)_h$ and thus, will have to use interactions to update $((\mathcal{M}^R)_h)_r$. In [11], authors provide explanations generated using model differences between $\mathcal{M}^R$ and $((\mathcal{M}^R)_h)_r$, show them to a human and let them accept or reject explanations. If an explanation is rejected, they use it as a signal to update $((\mathcal{M}^R)_h)_r$ and thereby, $\mathcal{M}^R$. This results in the robot finally having to change the original plan it came up with. Another approach to pre-compute and thereby speed up the $\Delta R$ and $\Delta H$ generated via this iterative process would be to have conditional plans where if an explanation is not accepted by the human, a different plan is selected. Lastly, an approach like probing may also help when the robot asks questions– either directly or by coming up with a set of planning problems and plans & explanations corresponding to the actions that are likely to be used in actual settings.

## Human Teammate

In this scenario, human, rather than acting as a supervisor, is a team member that can execute plan actions in the environment. Thus, it can provide support to the robot in the search and rescue tasks by checking some rooms. As explained earlier, in this setting the teammates can have their own team models, i.e. understanding about their and their teammate's capabilities. The robot, with computational power on its side, uses its team model to come up with a joint plan. Note that in the joint plan setting, there is an extra complication because the robot first needs to understand which models it needs to reconcile. In case the robot's part of the joint plan is inexplicable to the teammate, it must provide explanations that reconcile $\mathcal{M}^R$ and $((\mathcal{M}^R)_h)_r$. On the other hand, if

the inexplicability arises because the human is not sure why some of his actions are in the joint plan, it might have to provide explanations that reconcile $(\mathcal{M}^H)_r$ and $\mathcal{M}^H$.

**Robot knows better**   In this scenario, as before, the robot has the accurate model of its own ($\mathcal{M}^R$) and the human's capabilities (i.e. $(\mathcal{M}^H)_r$ is more accurate than $(\mathcal{M}^H)$). Thus, explanations generated by the robot reconcile $((\mathcal{M}^R)_h)_r$ to $\mathcal{M}^R$ and $\mathcal{M}^H$ to $(\mathcal{M}^H)_r$.

$\diamond$ **Use case** – Let's assume that robot knows that (1) there is fire in room 6 and (2) the human teammate cannot search a room that is on fire. Thus, it creates a plan in which the robot is supposed to check room 3, 5, 6 and 7 while human is supposed to check 1, 2, 4 and 8. The human teammate might not understand that why the joint plan does not involve checking consecutive room numbers. Robot needs to explain that there is fire in room 6, it has capability to look inside rooms on fire (reconciling $((\mathcal{M}^R)_h)_r$ and $\mathcal{M}^R$) and the human's lack of fire suit will not allow them to investigate a room that is on fire (reconciling $\mathcal{M}^H$ to $(\mathcal{M}^H)_r$).

$\diamond$ **Challenges** – As explained earlier, first, the robot needs to figure out which models is to be reconciled. In the worst case, as shown above, it might have to reconcile both for the current joint plan to make sense to the human. Secondly, in case the robot needs to reconcile $\mathcal{M}^H$ to $(\mathcal{M}^H)_r$, then it *has* to interact with the human as opposed to assuming $(\mathcal{M}^H)_r$ is an accurate representation of $\mathcal{M}^H$ (in contrast to previous work that assume $(\mathcal{M}^R)_h = ((\mathcal{M}^R)_h)_r$).

**Human knows better**   In this scenario, the human teammate has the accurate model of the task and thus, in this case, the plan based on $\langle (\mathcal{M}^R)_h, \mathcal{M}^H \rangle$ is valid and optimal and the plan the robot comes up with is therefore, inexplicable. Here the reconciliation can also be of two types– (1) $(\mathcal{M}^H)_r$ to $\mathcal{M}^H$ and (2) $\mathcal{M}^R$ to $(\mathcal{M}^R)_h$.

$\diamond$ **Use case** – In the previous scenario if we assume that human knows about the fire in room 6, then he will make the plan to go room 1, 2, 4 and 8 where as the robot would make the plan to visit room 1-4 by the robot and 5-8 by the human. Since human knows about fire in the room it can explain to the robot that room 6 can't be accessed by him, thus, leading to replanning scenarios to understand the model differences.

$\diamond$ **Challenges** – This scenario has three main challenges of which one is novel and two others have been discussed previously. First, the human's model $\mathcal{M}^H$ may be unknown and therefore, an accurate approximation of it in the robot's mind, i.e. $(\mathcal{M}^H)_r$ might not be available. Thus, the best thing to do for the robot would be to agree about achieving some goals/landmarks (or making other commitments about resources etc.) that it respects and expects the human to do the rest. A notion of net-benefit planning [12] may be useful is such settings. Second, determining which models to reconcile is a challenge similar to what was discussed in the previous case. Third, the idea of interaction in which the human teammate can come up with alternative foils and/or choose to not accept a particular explanation might trigger a conversation that updates the robot's set of models.

Although we discuss various ways to do model reconciliation, the idea of interaction based explanations seems to pop-up in many places. We thus, discuss this idea in a little more detain in the next section.

## Interaction-Based Model Reconciliation

Our central problem is to reconcile – (1) $(\mathcal{M}^R)_h$ with $\mathcal{M}^R$ and (2) $(\mathcal{M}^H)_r$ with $\mathcal{M}^H$ using the plans constructed by each teammate. The conversation has to be driven by the idea to reconcile models which may be annotated [10] or have some kind of abstractions like goal (through commitments and failures) [12] or action for reconciliation. Working on the more general problem of reconciling for joint plans, we will be able to effectively handle the human-as-a-supervisor scenario. We now present two different strategies for interaction that has not been discussed in earlier works i.e. to present foils or to probe for the model differences.

**Foil-Based Interaction**   Imagine the conversation between Robot (R) and Human (H):

```
R: How about the plan I go left and check
rooms 1-4 and you check rooms 5-8.
H: But you need to check rooms 3, 6, 7 & 8.
R: Why do you want me to check rooms 6, 7 & 8?
H: Room 6 and room 8 are on fire and I can't
search it. 7 might have caught fire too!
```

In this scenario, the human provides an alternative plan to the robot. On seeing this, the robot asks the human about the actions that the robot wanted the human to do. The human gives explanations that help the robot reconcile $(\mathcal{M}^H)_r$ to $\mathcal{M}^H$ (human can't search rooms that are on fire) and also update the task model $\mathcal{M}^R$ (if adjoining rooms are on fire, a room might catch fire).

**Probing**   On the other hand, imagine:

```
R: How about the plan I go left and check
rooms 1-4 and you check rooms 5-8.
H: Nope. That doesn't work.
R: Would you be fine if I check room 6, 7 & 8?
H: Ya. That works.
R: Did you say no to the first plan because
room 6 and room 8 are on fire?
H: Yes.
```

In this case, the robot inferred from the human's disagreement of the first plan that for some reason that the human does not want to search some of the rooms from 5-8. It suggested an alternative plan that if the human agrees to will help it easily verify its hypothesis that the human's search action has a precondition that the room is not on fire, thereby updating $(\mathcal{M}^H)_r$.

**Planning and Execution Phases**   The difference between planning and execution phase is about knowing the common plan $\pi$ as well as the goals which the team has agreed on in the execution phase. The conversation between the teammates would be driven to find the actions that are effected by a failing precondition or effect in the environment. We looked at the conversation strategies that can be followed for explanations through model reconciliation, we believe that conversation in both phases will be driven by different parts of the model to reconcile.

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
