# OpenReview forum: "Bidirectional Model Reconciliation: Explanations in Human-Robot Teams"
_icaps-conference.org/ICAPS/2019/Workshop/XAIP — Submitted to XAIP 2019_

### Official Review · AnonReviewer2 · 2019-05-09
**Clear description of challenges in model reconciliation, but lacking detail and placement wrt related work**

**Rating:** 1
**Confidence:** 2

**Review:**

The paper describes previous works in model reconciliation and discusses some related challenges. The paper succinctly and clearly describes the model reconciliation approach. I have two main concerns with the paper. The first is that it is not clear what is the contribution of this paper, and what is already included in the related works. While some challenges are clearly outlined, there is not enough detail in how these challenges might be approached or evaluated to form a significant enough contribution.

The second concern is that the challenges are not placed at all with respect to related work. [10/12] references in the paper are from the same group. I don't think that the authors should make the claim that the challenges proposed in the paper are unrelated to any other existing work, and that this avenue of research should be tackled in isolation.

The paper mentions foils and interacting through a dialog as approaches not before discussed in earlier works. In fact, there are many works that discuss this, in both XAI and XAI Planning. Two examples:
"David Smith; Planning as an Iterative Process; 2012" already discusses foil-based interaction through a dialogue for explanation.
"Miller. Contrastive explanation: A structural-model approach. 2018" and "Miller. Explanation in artificial intelligence: Insights from the social sciences. 2019" both describe the use of foils to form contrastive explanations.

Further, there are many works in human-robot interaction that also explore the idea of human-robot dialogue in human-robot teaming, which might present more realistic examples, approaches, and ideas for evaluation. Eg:
"Sebastiani et al. Dealing with On-Line Human-Robot Negotiations in Hierarchical Agent-based Task Planner. ICAPS 2017"

The previous works discussed in the "robot knows better" role make the assumption that the model (M^R)h is the same as ((M^R)h)r. In the case that they are different, I don't think this necessarily leads directly to "the human knows better". Obtaining an accurate representation of the human's mental model seems like the most difficult part of the process: is it possible that the representations (M^R)h and ((M^R)h)r differ without (M^H)r necessarily being correct?

What is the reason for having two models M^R and M^H rather than a single shared model, that jointly describes the robot's and human's capabilities? It appears from the discussion that a single initial state is shared (e.g. which rooms are on fire) and from this a single plan is generated containing actions from both human and robot. If this is the case, is the separation of M^R and M^H only on the set of actions A?
A. If not, how are plans generated and initial states reconciled - for example, reconciling M^H to (M^H)r does not account for the different model M^R.
B. If instead only the set of actions is partitioned, what is the advantage of reasoning about M^H and M^R separately? The reason for this is not clear in the paper.

In the introduction: "that is can use" - that it can use
In Use Cases: "respnses" - responses

---

### Official Review · AnonReviewer4 · 2019-05-13
**position paper extending model reconciliation paradigm**

**Rating:** 3
**Confidence:** 2

**Review:**

The paper extends the model reconciliation paradigm, specifically the previous work line by Kambhampati's group, to consider the case where the robot may have it wrong and thus a 2-way communication is required. The paper delineates relevant cases in this context, poutlines issues and possible solutions.

Overall, I find this fine for the purpose of discussion at the workshop.

My main concern with the paper is the way in which it disscusses, or rather *not* discusses, any other work in the area. One gets the impression the authors think they are alone in XAIP. This is inconsiderate in the best case, arrogant in the worst. I urge the authors to improve this, in the papoer and its preesentation, if accepted.

A technical note is that the recursive nature of "x's model of y's model of x's model" seems deeply related to epistemic logic/epistemic planning. It would seem that the entire problem of model reconciliation could be formulated as an epistemic planning problem, with additional predicates representing which literals occur in preconditions/effects/etc. It would be nice if the authors could comment on that.

---

> ### Comment · AnonReviewer4 · 2019-05-20
> **p.s.**
>
> In hindsight I would like to apologize for the tone of my comment reg related work. The authors failed to discuss related work. This may be a simple innocous mistake in the write-up. I should not have used the terms "inconsidrate" and "arrogant". I hope you accept my apologies.

---

### Official Review · AnonReviewer1 · 2019-05-13
**Unclear early paper that focuses too much on prior work**

**Rating:** 1
**Confidence:** 2

**Review:**

This paper discusses the topic of model reconciliation, attempting to extend it from use in a human supervisor-robot subordinate interact to be useful in a human teammate-robot teammate interaction. The paper has several major failings: first, it's very unclear about its motivations and intent; it completely forgoes any wrap-up that might shed light on what the authors thought they had accomplished. Second, it's very difficult to read from a technical perspective, relying heavily on notation that appears to be somewhat inconsistently used. Third, it focuses half of its length (two of four pages) on review of prior material. It satisfies neither as a summary of prior work nor discussion of an advancement. I think that this paper is trying to do too much in too little space. As a 4-page paper,  the authors would do well to focus on either a broad discussion of the bidirectional model reconciliation problem, with less detail; or, a discussion of primarily new material, forgoing the existing human supervisor case, with additional detail on the human teammate case.

Specific notes:
Description and asymmetry of scenarios 1 and 2 on page 1 are very confusing.
How is Delta R represented?
"Task Model" used on page 2 with no definition.
It's not clear in Human Supervisor\Robot Knows Better on page 3 why the objective is (apparently) to update MRhr rather than MH, if MR is correct, but MH is wrong.
In the Human Supervisor\Robot Knows Better use case, an explanation is described as updating MRhr, but it seems as if MRh would be more appropriate.
Human Supervisor\Human Knows Better: Why is human modelled as successfully updating MR, but robot only as successfully updating MRhr? This indicates that users are expected to already understand the system perfectly, whereas the opposite assumption is not made. You state this assumption relatively clearly later, but then claim that you use discussion to get around it, so why is it asymmetric?
Human Teammate: "with computational power on its side"? Are you trying to argue that the robot processor is faster than the human brain? This sounds controversial at best.
Human Teammate \ Robot Knows Better: "MHr is more accurate than MH" seems like apples and oranges. The robot understands the human's understanding of the world better than the human understands the world? How do you even compare that?
One alternate interpretation, that "MHr is a better model of MH than MH itself" is trivially false.
Another alternate interpretatin says that "MHr is a better model of the world than MH". This only says that MHr is a bad model of MH, though.

---

### Decision · Program_Chairs · 2019-05-15

**Decision:**

Reject

**Comment:**

Sorry, but given two strong reject votes, we decided to reject this paper. We hope the reviews give you useful information for revising your work. We wish you good luck with future iterations.